# Contextual constraints and dilemmas influencing health providers' prescription practices in a conflict-affected area: Qualitative insights from Mopti, Mali

Issa Coulibaly[1]*, Yacouba Diarra[1], Mohamed Ali Ag Ahmed[1,2,3], Raffaella Ravinetto[4,5], Seydou Doumbia[1], Karina Kielmann[4,6]

1 University of Sciences, Techniques, and Technologies of Bamako; Bamako, Mali, 2 Sherpa University Institute, Montreal, Canada, 3 Evaluation and Health Policy Department, University of Montreal, Montreal, Canada, 4 Department of Public Health, Institute of Tropical Medicine, Antwerp, Belgium, 5 School of Public Health, University of the Western Cape, Cape Town, South Africa, 6 Institute for Global Health & Development, Queen Margaret University, Edinburgh, Scotland, United Kingdom

* iss_coulibaly@yahoo.fr

## Abstract

Armed conflicts present complex, multidimensional challenges that severely compromise both access to and the quality of healthcare, including the adequate prescription of essential medicines. This study aimed to identify and understand the factors underlying the irrational prescribing of medicine in areas affected by armed conflict in the Mopti region in Mali. An exploratory qualitative study was conducted using a semi-structured interview guide to collect data from 30 participants, including health professionals from three health districts, representatives of the regional health directorate, members of community health associations, and staff from non-governmental organisations (NGOs) working in health sector. Interviews were recorded, transcribed, and subjected to thematic content analysis, using NVIVO 14 (2023 version) to support coding and data retrieval. Study participants reported a range of health system- and patient-related factors contributing to irrational prescribing practices in these conflict-affected areas. Health systems factors included: an imbalance between workload and the availability of qualified staff; limited access to professional training and training resources; poor adherence to prescription guidelines and procedures -often a consequence of contextual constraints; and coercion from armed terrorist groups. Administrative and political challenges were also highlighted, including weak monitoring and supervision mechanisms within the local health system; a lack of oversight by health and regulatory authorities in blockaded areas; the development of an informal pharmaceutical sector to compensate for deficiencies in the formal system; and overprescription of medicines linked to insurance schemes. Patient-related factors included delayed care-seeking linked to regional insecurity and contributing to increased morbidity and worsened health outcomes. Our results show that the

**Data availability statement:** The raw data from this study consists of interview transcripts containing potentially identifiable and sensitive information about participants. Sharing qualitative transcripts in open access would violate our ethical responsibility to ensure anonymity. Data are available upon request from the Ethics Committee of the Faculty of Medicine and Odontostomatology (FMOS) in Bamako, Mali, via telephone ((00223) 20 22 52 77), email (mdiakite@icermali.org), or through their website (https://fmos.usttb.edu.ml/) for researchers who meet the criteria for access to confidential data.

**Funding:** This study was financially supported by ELRHA (https://www.elrha.org), a UK-based humanitarian organization, through the Research for Health in Humanitarian Crises project, in the form of a grant (76507) (ELRHA-R2HC to IC, ELRHA-R2HC to MAA, ELRHA-R2HC to RR, ELRHA-R2HC to KK). No additional external funding was received for this study. The funder had no role in study design, data collection and analysis, decision to publish, or preparation of the manuscript.

**Competing interests:** The authors have declared that no competing interests exist.

multiple contextual factors influencing prescribing of medicines are deeply inter-linked. Efforts to improve the quality of prescribing in conflict zones must recognise that practices often deemed 'irrational' are shaped by complex contextual constraints. A coordinated and comprehensive approach involving all health systems stakeholders is required.

## Introduction

Health and health care access is often severely compromised in countries experiencing armed conflict. Health systems are weakened, the health infrastructure is damaged, and access to medicines is limited [1], resulting in reduced availability and quality of healthcare services, and low uptake of services even when available [2]. The ensuing fragility of the health system and reduced or sporadic availability of medicines can increase the demand for essential medicines, whilst simultaneously fostering an environment that favours sub-standard prescribing practices [3–5].

According to the World Health Organization (WHO), more than half of medicines prescribed globally are inappropriately dispensed or sold, and half of all patients do not take prescribed medicines correctly [1,6]. Irrational prescribing is one aspect of a broader concept of the inappropriate use of medicines, and refers to the prescription of incorrect medicines, unnecessary prescriptions of expensive medicines, or medicines prescribed with incorrect duration, route of administration, or dosage. More generally, it encompasses prescribing practices that do not align with the relevant national or international guidelines [2,3,7]. Irrational prescribing manifests in various forms, including polypharmacy, the inappropriate use of antimicrobials [3], and excessive prescription of injectable medicines [1,4]. The consequences of such practices include sub-optimal – or absent - therapeutic effects, adverse drug reactions and unfavourable cost-benefit ratios, all of which contribute to poor health outcomes and unnecessarily high health care expenditures [2,7].

A recent systematic review suggests that factors underlying irrational prescribing can be categorised under patient-related, provider-related, administrative and political issues [8]. Patient-related factors include delays in care-seeking that arise as a result of financial insecurity, non-adherence to prescribed treatments, and treatment-seeking in the informal sector [7,9]. Provider-related factors linked to irrational prescribing are reported to include medical paternalism, poor communication with patients, the multiplicity of prescribers and low levels of knowledge. Administrative and political issues include the high cost of medicines that can deter timely care-seeking; insurance incentives that may encourage polypharmacy; and challenges in monitoring and supervision in healthcare establishments [8].

Despite significant progress made in raising awareness of the rational use of medicines, training healthcare professionals, and improving distribution systems, irrational prescribing remains frequently reported in many low- and middle-income countries (LMIC) [10], particularly in poorly regulated health systems [7]. For example, irrational prescription and use of antimicrobials are major contributors to antimicrobial

resistance (AMR) [7,11,12]. In countries affected by armed conflict, health systems experience exceptional disruption, exacerbating the problem of irrational prescribing [13]. This has been highlighted in several studies, conducted in Yemen, Colombia, Syria, Côte d'Ivoire, the Democratic Republic of Congo (DRC), Sudan, and Nigeria, and which document the negative impact of conflict on the rational use of medicines [10–12,14–17]: the disruption of the supply chains of essential medicines severely limits their availability and access for affected populations.

Untrained or poorly qualified health staff - working in an unregulated context - are more likely to engage in irrational prescribing, which can increase medicine safety risks, pharmacological resistance, and poor health outcomes [12,16,18–20].

Mali, like other Sahelian countries, has been experiencing an unprecedented, multidimensional crisis for more than a decade. Marked by chronic instability due to armed conflict, socio-political tensions and the growing impact of climate change, the country is in a state of economic, social and institutional fragility [20,21]. This situation has weakened the local health system and increased the risks of irrational prescribing, as documented in a number of quantitative studies [22–24]. However, there remain significant gaps in understanding the factors associated with irrational prescribing in this context, where access to healthcare and the use of essential medicines have been severely impacted [16]. A recent study conducted by our team in conflict-affected areas of the country indicates sub-optimal performance on all indicators commonly used to assess irrational prescribing [25].

## Methodology

### Ethics statement

Ethical approval was obtained from the Ethics Committee of the Université des Sciences, des Techniques et des Technologies de Bamako (USTTB) in Mali (N°2023/182/CE/USTTB) and from the Institutional Review Board (IRB) of the Institute of Tropical Medicine in Antwerp, Belgium (N°1664/23). Written consent was obtained from each study participant prior to conducting and recording the interview.

### Study setting

Mali is a country in West Africa with a population of approximately 22,395,000 (as of 2022) and a land area of around 1,240,000 km$^2$. Mali's healthcare system is organised in a pyramidal structure, comprising three levels: central, regional, and operational. The operational level includes community health centres (CSCOM) which provide primary care, and reference health centres (CSREFs) which manage more complex cases. The private sector and NGOs also play a crucial role in healthcare provision. However, the system faces considerable challenges, including inadequate infrastructure, staff shortages, conflict-related instability, and insufficient funding [26].

The study investigated prescribing practices in the Mopti region, through interviews with stakeholders at the operational level in Mopti, and the central level in Bamako. Located in central Mali, Mopti is the country's 5th largest administrative region, with a population of around 2,570,513 and a surface area of 79,017 km$^2$ [27]. This region was selected for the study because it lies at the epicentre of the conflict yet was accessible at the time of data collection. Further, the research team had existing contacts with key stakeholders who could facilitate access. In addition to its indigenous population, Mopti hosts thousands of internally displaced persons and refugees from Burkina Faso. The region is organised into eight health districts. Data for the study was collected in three randomly selected health districts -Bandiagara, Djenné and Koro - and at the regional level in Mopti. Bamako was included as it houses the central services of the national health system, and the headquarters of several national and international organisations.

### Study design

We conducted a cross-sectional, observational study using qualitative semi-structured interviews. Data collection took place between September and November 2023.

## Study population and sampling strategy

Thirty individuals were selected from across the three levels of the health systems pyramid (Table 1). Informants representing the central level (n = 4) had a broader understanding and experience of the national pharmaceutical system and its regulation. Two officials were selected as informants from the regional level (n = 2). Finally, the largest group of informants comprised front-line service providers (n = 24), including doctors, nurses, pharmacists and sales depot managers.

## Profile of participants

Our sampling strategy was purposive and guided by the criteria of maximum diversity as well as saturation [28–31]. The doctors (n = 12) were all male, aged between 30 and 57 years, with professional experience ranging from 4 to 30 years. The pharmacist (n = 1) was also male, aged 43, with 13 years of experience in the field. The nurses (n = 10), all male, were aged between 26 and 43 years, with between 5 and 23 years of professional experience. Among the managers (n = 6), three were women, aged between 28 and 56 years, with between 5 and 23 years of experience. Finally, one participant representing the Association de Santé Communautaire (ASACO) was a 60-year-old man with 13 years of experience in this role.

## Participant recruitment

Administrative permissions for participant recruitment were obtained from the central and regional authorities. A three-day national workshop was held in Bamako in August 2023, during which we presented the research project to key stakeholders in the pharmaceutical sector. Following the workshop, a one-day meeting was held with regional stakeholders. These two meetings enabled us to identify and list potential participants at each level (operational, regional and central). We sought to include only adults aged 18 or over that fit into the professional categories listed above, were French-speaking, available and willing to participate and to provide informed consent. Appointments were made with those individuals who expressed interest to explain the study aims in detail, confirm eligibility, address any arising questions, and seek consent for the interview.

## Data collection

A topic guide was developed based on existing literature and our experience conducting a quantitative survey on prescribing practices [24]. The guide was structured to explore patient- and provider-related factors, as well as administrative and political issues, that might influence prescribing practices. The topic guide was pre-tested with a medical doctor and a pharmacist in a health district 15 km from Bamako and with a senior health technician working in a CSCom in Mopti. Modifications to the guide following the pre-test included getting rid of redundant questions and revising the order of several questions regarding drug management.

**Table 1. Breakdown of participants by profession and structure.**

| Participant profile | Central departments of public services/NGOs | Regional health directorate (DRS) | Reference health centres (CSRéf) | Community health centres (CSCom) | Humanitarian actors | Total |
|---|---|---|---|---|---|---|
| Doctors | 2 | 2 | 6 | 1 | 1 | 12 |
| Pharmacists | 1 | | | | | 1 |
| Nurse | 1 | | | 8 | 1 | 10 |
| Managers | | | 1 | 5 | | 6 |
| Members of community health association | | | | 1 | | 1 |
| Total | **4** | **2** | **7** | **15** | **2** | **30** |

The final guide covered the following topics: professional background; regulatory, socio-economic and systemic factors influencing use of medicines; prescribing practices; and factors that promote or hinder irrational prescribing. Following agreement on the time and place of the interviews with participants, face-to-face interviews lasting between one to one and a half hours were conducted in French, by an interviewer experienced in qualitative research and supported by a trained research assistant. All interviews were conducted at participants' workplaces and recorded with the participants' consent. Interviews were transcribed by the research assistant and subsequently cross-checked by the lead interviewer to identify any passages that might have been incorrectly transcribed or omitted during the process.

## Data processing and analysis

Interview transcripts were analysed using thematic categorical analysis [32]: first, data were organized into themes or categories (vertical analysis); second, emerging themes or categories were compared across interviews to identify trends or differences (horizontal analysis) [33]. Categorization (i.e., coding of the text according to the selected themes), inference (i.e., explaining what led the participants to make certain statements), and interpretation (i.e., drawing implications for our research questions) were the key stages in our analysis. We used the qualitative data analysis software NVIVO 14 as it allowed us to structure the data based on a hierarchical index of categories (referred to as 'nodes'), segment the interview transcripts and categorize the responses according to the themes we identified, enabling interpretations relevant to our research questions [34].

## Results

### Health systems challenges in the context of conflict

In conflict-affected areas, systemic weaknesses may directly impact on the behaviour of healthcare providers, forcing them to shorten consultations and prescribe quickly. As one of the doctors noted:

*"We are inundated with patients daily, and there are simply not enough of us to meet all the medical needs. I'm the only doctor here... I can't take the time needed with each patient."* (Doctor, operational level).

The constant pressure on medical teams undermines the quality of care and encourages hasty, sometimes inappropriate, prescriptions that respond more to the situation's urgency than to actual clinical needs.

The pressure to respond quickly is exacerbated by constant insecurity and the threat of violence. In this context, healthcare workers have to make quick clinical decisions - in some cases, without adequate information - leading to the increased risk of errors or inappropriate prescriptions. Medical history-taking is often abbreviated and discussions with patients cursory. A doctor commented on the impact of insecurity on his and his colleagues' practice:

*"The fear of armed groups and the urgency to treat as many patients as possible makes us cut corners. We don't have the luxury of taking a thorough medical history or explaining treatment options to the patients. Everything is rushed."* (Doctor, operational level).

Overall, the scarcity of resources, time constraints, and security threats foster an environment in which health care delivery is compromised, and suboptimal prescribing becomes a coping strategy rather than an instance of malpractice.

### Health care worker-related challenges

Study participants reported a series of health care worker-related issues contributing to irrational prescribing including a serious imbalance between the availability of qualified staff and the workload, and health care workers' limited access to professional training and training resources.

**Workload and lack of qualified staff.** The mass departure of healthcare professionals has reduced the number of available staff, leading to rushed consultations and compromised quality of care. In extreme cases observed, just two qualified staff members were available per 100,000 inhabitants, far below recommended standards. Describing the consequences of an imbalance between high patient loads and insufficient staffing, one doctor who reported seeking 70–80 patients and covering both the physicians and the maternity ward noted that he was unable to spend time with patients or provide them with the attention they required.

In this context, inappropriate prescribing is less a matter of individual error than a reflection of systemic breakdown. The combination of staff shortages, insecurity, and patient overload forces providers to prioritise speed over quality, highlighting the need for structural solutions to restore safe prescribing practices.

**Restricted access to training and medical resources.** The lack of continuing education has often been cited as an obstacle to appropriate prescribing. Without regular updates to their knowledge, healthcare professionals mainly referred to the national list of essential medicines, which was often cited as the only reliable source of guidance in a context of limited resources. A nurse testified to this gap, noting:

*"We are cruelly lacking in in-service training here. Our reference is the national list of essential medicines because the State has chosen to promote them, and we support them in this health policy." (*Nurse*).*

This situation was exacerbated by the lack of support tools. Manuals and guidelines available are often outdated or unsuited to the realities on the ground, limiting healthcare workers' ability to make informed decisions.

Although these limitations are experienced at the individual level, they reflect deeper systemic dysfunctions. The weakened health system no longer has the capacity to provide its workforce with essential resources — both technical and pedagogical — needed to maintain prescribing practices aligned with best clinical standards..

**Poor application of guidelines.** In conflict zones, several participants noted that there was lower adherence to medical protocols than in non-conflict situations. The notion of 'freedom to prescribe' was frequently invoked, particularly by doctors, but perceived by some as a pretext for bypassing evidence-based guidelines. One doctor, at the central level, commented that members of his profession staunchly believed that their freedom to prescribe was a "absolute right" seemingly in defiance of the legal, policy, and ethical codes around medical practice.

Other participants pointed out that non-compliance with guidelines was not solely a matter of individual choice, but often resulted from the structural constraints discussed above, in which time pressure and lacking resources led to a search for quick solutions:

*"There's a tendency to prescribe whatever seems necessary without following the guidelines, because the situation is urgent, and there is pressure to act fast. Often, these decisions are not based on the patient's needs but on what is available or what we think is the quickest solution."* (Doctor, operational level).

**Armed groups' pressure on health care workers.** Several professionals reported threats and acts of intimidation that forced them to rush care, often in a climate of fear, compromising the quality and rigour of prescriptions:

The generalised climate of insecurity fuels a constant sense of fear among both healthcare providers and patients, with strong impact for them at personal and psychological level (besides the impact on the health system). This collective anxiety leads to rushed medical consultations: doctors fear reprisals, and patients are anxious. This dynamic contributes directly to inappropriate prescriptions, often issued without sufficient explanation, as noted by one of the nurses interviewed:

*"Patients from insecure areas are in a hurry to get home. The prescriber knows that they come from insecure areas. If he takes enough time with them, their lives will often be in danger. […] This is one of the factors that explains the speed of prescriptions in these areas. "* (Nurse, Technical and Clinical Director).

The mass exodus of qualified personnel from areas under the control of armed groups has led to task-shifting to agents, for example, care assistants, who may lack the training or technical skills necessary to provide quality care, as explained by one of the doctors:

" *Care assistants have acquired some experience from the heads they were assisting. But that doesn't give them the technical ability to prescribe like someone who has undergone technical training.* […] *As a result of the crisis, people have left and the orderlies there have taken up the post with very limited skills.*" (Doctor, operational level)

**Administrative and political factors influencing IPM**

Several administrative and political issues contribute indirectly to the phenomenon of irrational prescribing in these conflict-affected areas. They include the absence or inadequacy of monitoring and supervision mechanisms, the unintended consequences of humanitarian assistance, the expansion of the informal sector; and overprescription, often triggered by incentives from the health insurance systems.

*Lack of adequate monitoring and supervision of activities.* The lack of effective oversight and monitoring mechanisms, largely attributable to the conflict situation, has encouraged the emergence of deviant practices in the health sector. In the absence of audits, some community health centres have taken advantage of the situation to increase the price of medicines. One doctor noted that in some of the health centres:

" … *the price of medicines has risen to between 3,500 and 5,000 CFA francs, almost to the level of prices in the private sector, where they vary between 6,000 and 7,000 CFA francs*". (Doctor, central level)

This sentiment was echoed by several other participants, reflecting a widespread concern that the lack of oversight contributed to a worsening of prescribing practices, due to poor accessibility of first-time medicines.

**Humanitarian assistance and its unintended consequences.** Humanitarian intervention, though essential for addressing the urgent needs of populations in crisis, can sometimes inadvertently encourage inappropriate prescribing practices. In some cases, the free and widespread distribution of medicines by humanitarian partners, without sufficient supervision, was reported to lead to their excessive or inappropriate use within local healthcare facilities. One regional-level doctor commented on the negative (unforeseen) consequences of humanitarian medicine donations:

"*Some* [humanitarian partners] *come with all the medicines for a minimum basket of activities, which they make available to health facilities to treat people affected by the crisis. The availability and self-service of these medicines leads to their abuse during the crisis.*" (Doctor, regional level)

Several other participants similarly observed that the ease of access to medicines without adequate oversight or rationing often led to their overuse or misuse, further exacerbating the problem of irrational prescribing in the affected regions.

**Development of the informal sector.** The collapse of regulatory mechanisms has led to the emergence of an informal market for medicines, which is particularly visible in areas under armed control. Unauthorised agents, often former healthcare professionals, prescribe and sell medicines outside official channels, thereby increasing the risks associated with their unregulated use. A doctor at the operational level described the proliferation of informal health actors and the consequences for patients' medicines purchases:

"*Hygienists and street vendors, often unemployed nurses or former health facility employees, use their stock to sell medicines. For example, they prescribe products that are not available from the official sales depot, forcing patients to buy these medicines directly, contributing to their unregulated use.*" (Doctor, operational level)

The lack of regulation of the informal medical market can lead to inappropriate prescriptions, in turn resulting in sub-optimal treatment and drug resistance. This issue also presents a challenge to formal healthcare structures, which struggle to compete with the lower costs of medicines sold by unregulated vendors. Several study participants link this situation with the weakness and shortages found in the formal pharmaceutical sector.

**Overprescription linked to health insurance's incentives.** Although health insurance mechanisms are intended to improve access to healthcare, it was reported in this context that they can also act as perverse incentives. Several participants reported situations where providers took advantage of third-party payments to prescribe unnecessary medication, hoping to receive reimbursement without rigorous checks. These incentives contributed to an increase in irrational prescriptions, as described by one of the regional-level doctors:

*"The presence of the insurance company means that some prescribers abuse their prescriptions. Under the agreement that binds us, we pay for the service for our policyholders. In other words, once the treatment has been paid for, we reimburse the approved centers monthly. What happens? As we are reimbursed, some prescribers sometimes prescribe a lot of drugs to insured people in the hope that we won't see them and that we'll reimburse everything."* (Doctor, regional level).

This highlights how financial incentives, when not adequately regulated, can distort prescribing behaviours.

## Patient-related factors influencing irrational prescribing

The two main patient-related factors influencing irrational prescribing behaviour were patients' delayed care-seeking and the influence of their preferences and demands on prescribing behaviour.

*Delayed care-seeking.* In conflict-affected areas, insecurity presents significant challenges for access to health care leading to delays in timely care-seeking that potentially result in a cumulative illness burden, and consequently, an increase in the number of medications prescribed. The occurrence of multiple untreated co-morbidities demanded more complex and costly treatments, as reflected in the comments of one of the doctors interviewed:

*"Insecurity means that patients don't get to the centre on time. They may have several morbidities at the same time. For example, a patient with a urinary infection who doesn't come in time. Then, if malaria is added, the patient will be treated differently from a patient with a simple urinary infection who is easy to treat. The fact that people don't come in time may factor in irrational use. Instead of prescribing for 2,000 francs, we'll go up to 5,000 francs …."* (Doctor, operational level).

These concerns were widely shared among participants: the presence of multiple morbidities that had to be treated could result in polypharmacy, with unanticipated side effects resulting from multiple pharmacological interactions.

**Patient preferences.** In some health districts, particularly Bandiagara and Koro, healthcare providers reported that patients expressed strong preference for injectable medicines which were perceived as more efficacious than in oral form. Faced with the prospect of poor compliance, some providers adapted their prescriptions in favour of injectables, even for simple cases where these were not indicated. These decisions, highly influenced by local preferences, were seen to contribute to the inappropriate use of medicines. A nurse provided an example of community-level resistance to 'simple' treatment for malaria:

*"In our district, even if we were in Koro health district you won't find anyone who believes that arthemeter* [derivative of artemisinin]*, for example, should only be prescribed for severe malaria cases.* […] *when I came to this post, I gave them* [the community] *tablets for all the simple cases. What I notice when I go home is that I see my tablets by the*

*roadside or in the water, and what I hear in the village is, 'He only prescribed tablets, but I've taken enough.' So, if you're in an area like that, you must change".* (Nurse)

Additionally, due to the departure of many prescribers at the start of the conflict, hygiene and sanitation workers became the de facto prescribers. As they were not qualified to prescribe treatment, they ended up performing medical procedures, with potentially serious negative consequences. Hygienists were also reported to overprescribe injectables for financial reasons, as patients could be charged for having injections at home.

**Patient non-adherence.** Patients' non-adherence to prescriptions has prompted some providers to adapt their prescribing practices. As consultations often take place on market days, health care professionals anticipate the difficulties that may arise in monitoring patients by prescribing more medicines than strictly necessary, or medicines and injectables perceived to be more potent in order to ensure patient compliance. Providers recognise that patients travel far to access a provider, even if the problem is small, and have high expectations. One doctor aptly describes the dilemma arising when patients are dissatisfied with a generic medicine:

" *You realise that the problem isn't that big and give them the medicine in its International Non-proprietary Name (INN), especially the bulk form. He* [the patient] *will tell you that in their CSCOM, the medicines are the same. We know that in treatment, there is what is known as the placebo effect. Doctors don't count this effect. However, what is certain is that if the patient is not satisfied with the quality of the medicine when they leave, they will not take it. For someone who has travelled from 100 km away, if we prescribe the INN, he won't take it. Sometimes, this leads us to prescribe injectables or antibiotics".* (Doctor, operational level)

In this situation, what appears to be 'irrational' prescribing is in fact a pragmatic and adaptive response on the part of the provider to the realities of care delivery in insecure and remote settings.

## Discussion

Offering unique insights on factors associated with irrational prescription of essential medicines in a conflict-affected areas in Mali, this study complements previous work in the area [35] highlighting factors associated with irrational prescribing linked to providers [36], administrative and political issues [37], and to patients. As previously reported, these factors are exacerbated by the conflict [38] that has further weakened an already fragile health- and pharmaceutical system.

We note the striking imbalance between high workloads and the lack of qualified staff which contributes significantly to irrational prescribing. Outnumbered and working under the pervasive fear for both their own safety and that of their patients, healthcare workers are unable to allocate adequate time to individual consultations. Consequently, the limited time available for prescribing and dispensing medicines contributes to the over-prescription, including unnecessary antibiotics and injectable treatments—an observation consistent with findings from similar contexts. For example, a study from Syria documented how the reduction in number of healthcare professionals during the conflict led to an increase in medical errors and irrational prescribing [39]. In the Democratic Republic of Congo (DRC), health care workers were often overworked; limited time and resources resulted in overly hasty prescription of medicines [40]. In Côte d'Ivoire, health care workers reportedly abandoned their posts for safer areas due to fear [15], leading to an increased burden of responsibilities for those health care workers who remained.

Lack of professional training and limited access to medical resources were also seen to be important challenges contributing to inappropriate prescribing practices. The few training courses available to health care workers in conflict-affected areas rarely focus on the rational prescribing of medicines [24]; previous studies have shown that armed conflict systematically disrupts not only healthcare services but training and support programmes for health care workers. In Yemen, for example, the interruption of medical training programmes due to the armed conflict led to insufficient

knowledge of recommended treatments, exacerbating the problem of irrational prescribing [41]. In Pakistan's conflict-affected areas, prescribers' lack of knowledge were reported to contribute to irrational antibiotic prescribing [11]. In Somalia, the lack of access to continuing education for health care workers led to incorrect prescribing and inappropriate disease management [42]; similar gaps have been documented in health care worker knowledge of adequate prescribing of essential medicines in Côte d'Ivoire [17], the DRC [16], Burkina Faso [7,12] and other LMICs [11,15,43,44], highlighting the urgent need for improved access to continuing medical education through channels accessible in unstable environments.

Our study indicates that non-adherence to medical guidelines is a significant problem in conflict-affected areas, resulting in irrational use of medicines. This phenomenon has been attributed to a lack of supervision and a lax interpretation of medical standards [45]. A study from Lebanon revealed that the lack of supervision and regulation in refugee camps led to widespread divergence from medical protocols, exacerbating prescribing problems and their consequences for individuals and communities [46]. Similarly, the absence of oversight of medical practices in Afghanistan resulted in significant variability in the quality of care and prescriptions [47]. This evidence highlights the need to strengthen supervision and regulatory mechanisms to reduce irrational prescribing practices. However, as we have noted above, what is labelled as 'irrational prescribing' must be situated within the contextual constraints present in conflict-affected areas. Indeed, the term 'irrational prescribing' masks the pressing social and contextual realities that shape providers' and prescribers' decision-making and agency in in conflict-affected areas.

Pressure from armed groups on health care workers represented an additional and complex challenge reported in the study, echoing findings from work conducted in Central African Republic [48] and Iraq [49] that similarly highlight the impact of security threats on the quality of prescriptions. There is converging evidence that pressure from armed groups can become a significant obstacle to the provision of rational and appropriate medical care in conflict zones.

In our study, we found several administrative and political factors contributing to irrational prescribing. Despite the dedication of health care workers continuing to deliver services in extreme circumstances, the health authorities' lack of monitoring capacity at different levels of the health pyramid contributes to irrational prescribing and higher expenses for medicines [3,8]. Similar findings were reported in a study from Yemen [50]. Further research will be needed to understand the impact of irrational prescribing on health-related expenses in conflict areas, at both the household and health systems levels, and to inform corrective actions [50].

The observed development of an informal pharmaceutical sector that has flourished in the face of a fragile formal sector is seen in other contexts. For instance, in the DRC, the proliferation of non-compliant medicines and unsafe prescribing practices in the informal medicines market was exacerbated by the lack of regulation [32]. A study in South Sudan revealed that the lack of supervision had allowed informal actors to replace health care workers, increasing the risks associated with the irrational prescription of medicines [51]. This evidence underlines the importance of rigorous monitoring and regulation of the medicines sector to ensure access to safe and effective care in the formal sector. However, the absence of qualified healthcare personnel and the frequent stock-outs of medicines in the public health structures has created a vacuum that generates a constant "demand" for care, providing an opportunity for rapid development of the informal sector.

We noted that some health care workers were reported to take advantage of incentives related to health insurance schemes for personal financial gain, through overprescribing or prescribing inappropriate but more expensive medicines. This was similarly reported from a study in South Sudan and Sudan, where corruption and [inadequate?] financial incentives had a negative influence on medical practices [52]: health care workers sometimes resorted to prescribing unnecessary treatments to maximise reimbursements.

Delayed health care-seeking is a frequent problem in conflict zones, linked to the insecurity: patient may even put their lives at risk to reach health facilities, thus complicating the management of illnesses diagnosed at an advanced stage, that may require more complex treatments and more intensive prescription of medicines (polypharmacy). Similar findings were

reported in the DRC and Colombia [15,16], however, these studies focused on financial constraints limiting or delaying people's access to health facilities in conflict zones [15,16].

We also identified the influence of patients' preferences on provider decision-making related to prescribing. Patients' preferences for injectables may stem from cultural considerations or personal beliefs regarding the efficacy of specific therapeutic modes [53] and motivate prescribers to deviate from established medical protocols [54]. Moreover, health care providers in conflict areas are under time pressures and unable to engage with patients. Further, logistical challenges, such as long distances to access care, contribute to non-compliance with prescriptions [55], even more pronounced in conflict areas. The preference for certain medications is influenced by regional insecurity, with long-acting injectables preferred over oral formulations. Additionally, a patient's financial situation or insurance coverage can impact prescriptions, as prescribers often accommodate patients' requests to avoid non-compliance or address their vulnerability. Similar observations were reported in the Palestinian conflict zone [56], where the main factors shaping patient preference were poverty, the burden of health insurance and mobility difficulties [56].

Notably, participants did not raise concerns about possible financial interests of the private pharmaceutical sector - whether multinational corporations or local wholesalers and distributors - shaping prescribing practices. This absence may seem surprising, given the well-documented impact of pharmaceutical marketing practices on prescribing behaviours and the rational use of medicines globally [57,58]. A possible explanation lies in the context of the study area, which is marked by high levels of insecurity and widespread poverty, and thus less attractive to commercial actors. However, this situation may change: as the acute effects of the conflict subside, it will be important to (re)examine the national and local governance of pharmaceutical promotion, as has been done in other sub-Saharan African settings [59].

In light of our findings, several actions are needed to address the reported *irrational* prescribing practices in conflict-affected areas of Mali, and possibly in comparable settings. It is critical to address the existing staffing shortages, gaps in training, and deficiencies in supervision of prescribing practices. On the administrative and political fronts, effective monitoring needs to be re-established to oversee the formal sector and to monitor informal sector prescriptions. Finally, patients need to be involved as partners in the prescribing decisions and explained the rationale for best (prescribing) practices rather having them imposed. However, none of these actions can deliver improvements if local health and pharmaceutical systems are not strengthened, and if essential medicines are not made available and affordable for all in the formal sector.

## Conclusion

By interviewing health care workers and policy-makers at different levels of the Mali health pyramid, various interrelated factors were identified as being at the root of inadequate prescribing of medicines. Poor prescription practices were either pre-existing and exacerbated by the conflict or were caused by the conflict. However, it is essential to recognise that inadequate practices that could easily be labeled as 'irrational' are often shaped by contextual constraints and ensuing dilemmas arising for health care providers. Under these circumstances, the use of the traditional wording 'irrational prescribing' seems inadequate, as it does not reflect the impact of social and contextual determinants on prescribing behavior, particularly within the complex setting of a health system affected by conflicts.

## Acknowledgments

We want to thank the members of the network Afrique Francophone et Fragilité (AFRAFRA); the Malian authorities at central, regional and operational levels; and the partner NGOs *Association Malienne Pour La Survie au Sahel (AMSS)* and *L'Alliance Médicale Contre le Paludisme-Santé Population (AMCP-SP)* for facilitating the realisation of this research project.

## Author contributions

**Conceptualization:** Issa COULIBALY, Mohamed Ali Ag Ahmed, Raffaella Ravinetto, Karina Kielmann.

**Data curation:** Issa COULIBALY, Yacouba Diarra, Mohamed Ali Ag Ahmed.

**Formal analysis:** Issa COULIBALY, Yacouba Diarra, Mohamed Ali Ag Ahmed.

**Funding acquisition:** Issa COULIBALY, Mohamed Ali Ag Ahmed, Raffaella Ravinetto, Karina Kielmann.

**Investigation:** Issa COULIBALY, Yacouba Diarra, Mohamed Ali Ag Ahmed.

**Methodology:** Issa COULIBALY, Mohamed Ali Ag Ahmed, Raffaella Ravinetto, Karina Kielmann.

**Project administration:** Issa COULIBALY.

**Supervision:** Issa COULIBALY, Mohamed Ali Ag Ahmed, Seydou Doumbia.

**Validation:** Issa COULIBALY, Mohamed Ali Ag Ahmed.

**Visualization:** Issa COULIBALY, Mohamed Ali Ag Ahmed, Raffaella Ravinetto, Karina Kielmann.

**Writing – original draft:** Issa COULIBALY, Mohamed Ali Ag Ahmed.

**Writing – review & editing:** Issa COULIBALY, Yacouba Diarra, Mohamed Ali Ag Ahmed, Raffaella Ravinetto, Seydou Doumbia, Karina Kielmann.

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
