## [Decision Letter · Decision Letter 0]

18 Jun 2025

PGPH-D-25-00971

Contextual constraints and dilemmas influencing health providers’ irrational prescribing of medicines in conflict-affected areas: qualitative insights from Mopti, Mali

Dear Dr. COULIBALY,

Thank you for submitting your manuscript to PLOS Global Public Health. After careful consideration, we feel that it has merit but does not fully meet PLOS Global Public Health’s publication criteria as it currently stands. Therefore, we invite you to submit a revised version of the manuscript that addresses the points raised during the review process.

Editor comments: 

The paper offers important insights into the contextual factors contributing to inappropriate prescribing practices in conflict-affected settings, specifically Mopti, Mali. It makes a valuable contribution to the existing body of knowledge, especially given the scarcity of qualitative research on how conflict environments can exacerbate irrational prescribing, with serious implications for health outcomes in contexts of poverty and instability.

That said, there are several issues that need to be addressed to strengthen the manuscript and make it suitable for publication:

**Language, Grammar, and Writing Style:**The manuscript requires thorough proofreading to correct a range of grammatical and stylistic issues. I observed multiple instances of awkward phrasing, overly long sentences, inconsistent use of abbreviations, and problems with cohesion. The clarity and impact of the paper would greatly benefit from breaking down complex sentences and ensuring correct grammar throughout.**Title Accuracy:**The current title refers to “conflict-affected areas” (plural), which is misleading, as the study focuses solely on Mopti, Mali. Please consider revising the title to accurately reflect the geographic scope of the research.**Overlooked Dimension: Financial or Corporate Incentives:**A major limitation of the current analysis is the absence of discussion around financial or corporate interests—particularly the influence of pharmaceutical companies. There is substantial global literature linking pharmaceutical incentivization to inappropriate prescribing practices (e.g., polypharmacy). It would be important to address whether this issue arose in your data. Did any participants mention pharmaceutical influence, or how companies might operate in or exploit conflict settings? If this theme did not emerge, it would be helpful to acknowledge its absence and reflect on why that may be the case.**Narrative Framing of Results:**While the structure of the results section is appropriate, it is currently too reliant on quoted material with limited analytical interpretation. For an international audience, it is essential that you provide a clearer narrative voice to guide readers through the themes. Rather than long descriptive passages, I recommend integrating brief but thoughtful reflections that help contextualize and interpret the quotes. This will strengthen the coherence and analytical depth of the results.

Overall, this manuscript has strong potential and addresses an underexplored area of global health research. With careful revisions—particularly in language, analytic framing, and attention to omitted thematic areas—it could make a meaningful contribution to the field.

We look forward to receiving your revised manuscript.

Kind regards,

Naveed Noor, PhD

Academic Editor

Journal Requirements:

Reviewers' comments:

Reviewer's Responses to Questions

**Comments to the Author**

1. Does this manuscript meet PLOS Global Public Health’s publication criteria?

Reviewer #1: Yes

2. Has the statistical analysis been performed appropriately and rigorously?

Reviewer #1: N/A

3. Have the authors made all data underlying the findings in their manuscript fully available (please refer to the Data Availability Statement at the start of the manuscript PDF file)?

Reviewer #1: No

4. Is the manuscript presented in an intelligible fashion and written in standard English?

Reviewer #1: Yes

Reviewer #1: Well written manuscript. Appreciate the flowless translation from French. Excellent review of literature.

Abbreviations, such as IPM, NGO could be defined at first mention for ease of reading

A couple of questions: Lines

152 Reference

156 NGO crucial role in what?

207 Quantitative survey?

277 The sentence is not clear

333, 341 I see empty [...]. Not sure if this is intentional

598 To improve working conditions aside staff, training, supervision, and monitoring, you may want to ensure security

**Do you want your identity to be public for this peer review?** For information about this choice, including consent withdrawal, please see our Privacy Policy

Reviewer #1: **Yes: ** maria a afadapa

---

## [Editor Report · Decision Letter 1]

25 Sep 2025

Contextual constraints and dilemmas influencing health providers’ irrational prescribing of medicines in conflict-affected areas: qualitative insights from Mopti, Mali

PGPH-D-25-00971R1

Dear Dr Coulibaly, 

We are pleased to inform you that your manuscript 'Contextual constraints and dilemmas influencing health providers’ irrational prescribing of medicines in conflict-affected areas: qualitative insights from Mopti, Mali' has been provisionally accepted for publication in PLOS Global Public Health.

Best regards,

Naveed Noor, PhD

Academic Editor